# Effect of Freeze–Thaw Cycles on the Performance of Concrete Containing Water-Cooled and Air-Cooled Slag

**Seung-Tae Lee [1,\*], Se-Ho Park [1], Dong-Gyou Kim [2] and Jae-Mo Kang [2]**

[1] Department of Civil Engineering, Kunsan National University, Gunsan 54150, Korea; sekhok88@kunsan.ac.kr
[2] Underground Space Safety Center, Korea Institute of Civil Engineering and Building Technology, Goyang 10223, Korea; dgkim@kict.re.kr (D.-G.K.); jmkang@kict.re.kr (J.-M.K.)
\* Correspondence: stlee@kunsan.ac.kr; Tel.: +82-10-3666-9561

**Abstract:** An experimental study on the resistance of concrete containing air-cooled slag (AS) and water-cooled slag (WS) against freeze–thaw cycles was conducted. For comparison, the durability of ASTM Type I ordinary Portland cement (OPC) concrete exposed to the same freeze–thaw environment was examined. To evaluate the durability of concrete exposed to the freeze–thaw environment, an experiment was conducted according to ASTM C 666 procedure A. Furthermore, the relative dynamic modulus of elasticity, surface electrical resistivity, and compressive strength of concrete specimens were measured after exposing them to freeze–thaw cycles for a predetermined period, and the results were compared with those of OPC concrete. The relationship between the freeze and thaw resistances of concrete and the air-void system (spacing factor and specific surface area) was identified. Furthermore, the microstructure of concrete exposed to freeze–thaw cycles was observed using scanning electron microscopy to identify the interfacial transition zone, cracks, and micropores. Experimental results showed that the resistance of blended cement concrete containing WS and AS against freeze–thaw cycles was significantly higher than that of OPC concrete. The concrete in which 10% of OPC was replaced by AS exhibited a similar durability as that of the concrete in which 40% of OPC was replaced only by WS. Therefore, it is expected that blended cement concrete containing WS and AS based on an appropriate mix proportion design will exhibit excellent durability in regions experiencing freezing temperatures.

**Keywords:** water-cooled slag; air-cooled slag; concrete; freeze–thaw cycles; air-void system; microstructure



## 1. Introduction

Alongside steel, concrete is one of the most important materials in the construction industry, and in general, the durability of concrete structures degrades because of some environmental causes, such as freezing and thawing, steel reinforcement corrosion due to chloride attack and carbonation, and chemical erosion. As such, many studies have been conducted on these causes worldwide [1,2]. In particular, when porous media, such as concrete, are exposed to a freeze–thaw environment, the water in the pore solution is subjected to repeated freezing and expansion, resulting in expansive cracks in concrete because of the increase in the tensile stress caused by expansive pressure [3]. The mechanism of concrete deterioration due to freezing and thawing has been reported by many studies. The most dominant mechanism reported to date is the glue spall mechanism proposed by Valenza and Scherer [4], who reported that the tensile stress caused by the difference between the thermal expansion coefficients of ice (generated by freezing) and concrete leads to cracking.

In addition, many previous studies have reported that the freeze–thaw resistance of concrete is affected by the type and amount of binder. Furthermore, many studies have reported that the concrete containing mineral admixtures possesses excellent frost resistance [5–7]. In general, mineral admixtures used as binders for concrete include blast furnace slag, fly ash, silica fume, and metakaolin. Among them, blast furnace slag can be

classified into water-cooled slag (WS) and air-cooled slag (AS), according to its cooling method [8].

WS is commonly used as an admixture for concrete, and approximately 30% of its output is used as a concrete binder in South Korea. However, approximately 70% of AS is disposed, and the rest is mostly used as a roadbed material, material for embankment, and silicate fertilizers [1]. Several studies [9,10] have indicated that AS can reduce the hydration heat and improve the fluidity; owing to these characteristics, its applicability as a concrete admixture has attracted attention. It has been reported that AS can be used as a concrete binder because it is suitable for developing the latent hydraulic property with calcium hydroxide in concrete when used as a concrete admixture, because its chemical composition is similar to that of WS, and it possesses a high basicity [10,11]. In addition, it is known that cooled AS does not require any special stabilization treatment unlike steel furnace slag because F-CaO and F-MgO hardly exist in AS [1,5,9].

Based on the physical and chemical properties of AS, several studies have been conducted wherein AS was used as a concrete material. In particular, Mostafa et al. [10] studied the microstructure of the hardened cement matrix containing AS, and El-Didamony et al. [11] attempted to experimentally verify the usability of AS as a substitute for WS. Moreover, the applicability of AS as a concrete binder was investigated in terms of the strength, fluidity, and durability of concrete. On the contrary, Verian and Behnood [12] experimentally studied the freeze–thaw resistance and scaling of cement concrete pavements containing AS aggregates. Nevertheless, only a few studies have been conducted on the freeze–thaw resistance of concrete containing AS as a binder.

In this study, concrete was mixed using AS and WS as binders, and concrete specimens were fabricated to experimentally evaluate the freeze–thaw resistance of hardened concrete, as well as its mechanical performance, such as the compressive and flexural strengths and air-void system. In addition, the microstructure of concrete subjected to freeze–thaw cycles was investigated. The experimental results of this study could provide useful information for selecting optimal binders for concrete structures built in regions experiencing very low temperatures.

## 2. Experimental Procedure

### 2.1. Materials

Ordinary Portland cement (OPC) 43 grade (IS: 8112-1989) was used in this study. In addition, WS and AS, which are among the industrial by-products of Pohang Steelworks in South Korea, were used as binders for replacing OPC to fabricate binary blended and ternary blended concrete. Figure 1 shows the scanning electron microscopy (SEM) images of the binders used in this study. The SEM images of the binders were obtained using an XL-30 ESEM at the same magnification (×10,000). The shapes of the AS particles were found to be similar to those of WS, but their surfaces were rougher. Table 1 lists the chemical composition and physical properties of each binder—it is seen that WS and AS have similar chemical components and densities.

**Table 1.** Chemical composition and physical properties of binders.

| | Chemical Composition (wt. %) | | | | | | | Physical Properties | |
|---|---|---|---|---|---|---|---|---|---|
| | $SiO_2$ | $Al_2O_3$ | $Fe_2O_3$ | CaO | MgO | $SO_3$ | LOI | Density (g/cm$^3$) | Fineness (cm$^2$/g) |
| OPC | 19.8 | 4.8 | 3.1 | 61.5 | 2.9 | 2.8 | 2.96 | 3.15 | 3400 |
| WS | 31.7 | 14.5 | 0.4 | 41.7 | 5.4 | 2.1 | 2.6 | 2.9 | 4700 |
| AS | 30.8 | 12.1 | 0.71 | 49.7 | 2.71 | 1.75 | 2.2 | 2.9 | 4300 |

Crushed sand (S) was used as the fine aggregate, and crushed gravel with maximum sizes of 19 mm (G1) and 32 mm (G2), satisfying ASTM C33/M33-18 [13], were used as the coarse aggregates. Their physical properties are listed in Table 2. In addition, polycarbonic

acid-based superplasticizer (SP) and air-entraining agent (AEA) were used to secure an appropriate slump (60 ± 10 mm) and air content (6 ± 1%) for all concrete mixtures.

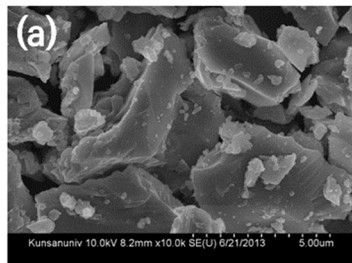 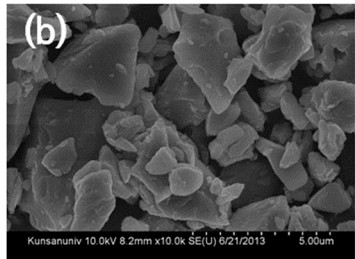 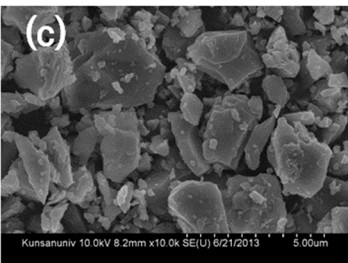

**Figure 1.** SEM images of the binders: (**a**) OPC, (**b**) WS, and (**c**) AS.

**Table 2.** Physical properties of aggregates.

|   | Gmax (mm) | Absorption (%) | Fineness Modulus | Density (g/cm$^3$) |
|---|---|---|---|---|
| S | - | 1.14 | 2.9 | 2.53 |
| G1 | 19 | 0.78 | 6.2 | 2.73 |
| G2 | 32 | 0.93 | 6.8 | 2.75 |

### 2.2. Mix Proportions

Table 3 shows the mix proportions of the concrete used in this study. The OPC concrete mix (reference mix) had a water-to-binder ratio of 45% and a fine aggregate ratio of 35%, while the unit binder content (C + WS + AS) was fixed at 350 kg/m$^3$. WS replaced 30%, 35%, and 40% of OPC, whereas AS replaced 0%, 5%, and 10% of OPC. After mixing, water curing was performed at 20 ± 3 °C for a suitable period for testing the mechanical performance and freeze–thaw resistance of concrete.

**Table 3.** Mix proportions of concrete (unit: kg/m$^3$).

| Mix. Code | W | C | WS | AS | S | G1 | G2 | SP * | AEA ** |
|---|---|---|---|---|---|---|---|---|---|
| OPC | 157 | 350 | - | - | 645 | 595 | 599 | 0.33 | 0.25 |
| WS40 | 157 | 210 | 140 | - | 632 | 583 | 588 | 0.3 | 0.4 |
| WS35AS05 | 157 | 210 | 122.5 | 17.5 | 639 | 590 | 594 | 0.5 | 0.5 |
| WS30AS10 | 157 | 210 | 105 | 35 | 661 | 609 | 614 | 0.55 | 0.5 |

* wt. of binder (%); ** wt. of SP (%).

### 2.3. Test Methods

The compressive and flexural strengths of the concrete subjected to water curing at 20 ± 3 °C were measured at 7, 28, and 91 days of age. The compressive strength was measured using three specimens (ϕ100 mm × 200 mm) per concrete mix according to ASTM C39/C39M-20 [14], i.e., Standard Test Method for Compressive Strength of Cylindrical Concrete Specimens. The flexural strength was measured using two concrete specimens (100 mm × 100 mm × 400 mm) per concrete mix according to ASTM C293/C293M-16 [15], i.e., Standard Test Method for Flexural Strength of Concrete.

The air-void system properties of hardened concrete (ϕ100 mm × 50 mm) at 28 days were measured according to the linear traverse method of ASTM C 457/C457M-16 [16]. A high-resolution flatbed scanner was used to acquire digital images of the concrete cross section. After polishing the cross section, a high-fineness white powder was applied, and then each pixel corresponding to the cement paste, aggregate, and air void was classified using the Image-Pro analysis software. Based on the image classification, the air content, specific surface area, and spacing factor of each concrete specimen were determined.

After fabricating prismatic specimens with the dimensions of 100 mm × 100 mm × 400 mm, a freeze–thaw resistance test was conducted according to ASTM C666/C666M-

15 [17] procedure A using specimens aged for 28 days considering the strength development of concrete. The 3 h-process, in which the standard specimen for temperature measurement is frozen for 1 h from 4 °C to −18 °C and melted for 2 h from −18 °C to 4 °C was set as one freeze–thaw cycle; a total of 300 cycles were performed. The dynamic modulus of elasticity of each concrete specimen was measured every 30 cycles (Figure 2). Then, the relative dynamic modulus of elasticity (RDME) of concrete was calculated as follows.

$$\text{RDME} = \left(\frac{n_c}{n_0}\right)^2 \times 100 \ (\%) \tag{1}$$

where $n_c$ denotes the fundamental transverse frequency after c cycles of freeze–thaw (Hz), and $n_0$ denotes the fundamental transverse frequency at the start of freeze–thaw (Hz). The durability factor (DF) of concrete was determined after 300 freeze–thaw cycles, as follows.

$$\text{DF} = \frac{\text{PN}}{300} \ (\%) \tag{2}$$

where P represents the RDME at N cycles, and N denotes the number of cycles where P is less than 60%; N is set to 300 when P is above 60% after the completion of freeze–thaw.

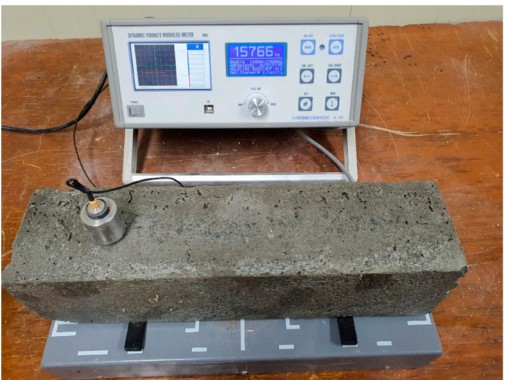

**Figure 2.** Measurement process of RDME.

To measure the surface electrical resistivity of the concrete subjected to freeze–thaw cycles, a four-electrode resistivity test was conducted according to the Wenner method (Figure 3) using cylindrical specimens (ϕ100 mm × 200 mm) after 0, 90, 210, and 300 freeze–thaw cycles. The non-destructive test was conducted using Resipod with four-point Wenner probe (Proceq). The surface electrical resistivity was calculated using Equation (3), and its decrease for each cycle from the initial value of concrete before exposure to freeze–thaw cycles was calculated.

$$\rho \ (\text{Kohm·cm}) = \frac{2\pi aV}{I} \tag{3}$$

where ρ, a, V, and I denote the surface electrical resistivity (Kohm·cm), distance between the electrodes (cm), potential difference (V), and current (A), respectively.

To evaluate the strength reduction of concrete due to freeze–thaw cycles, the compressive strength was measured according to ASTM C39/C39M-20 [14] after 0, 90, 210, and 300 cycles using cylindrical specimens (ϕ100 mm × 200 mm) subjected to standard curing for 28 days. The compressive strength loss (CSL) was calculated as follows.

$$\text{Compressive strength loss (CSL)} = \frac{(C_0 - C_c)}{C_0} \times 100(\%) \tag{4}$$

where $C_c$ denotes the compressive strength after c cycles of freeze–thaw (MPa), and $C_0$ denotes compressive strength before freeze–thaw (MPa).

To observe the microstructure of the concrete samples subjected to 210 freeze–thaw cycles, the surface areas of the samples were magnified appropriately (×5000) and observed using XL-30 ESEM (Philips).

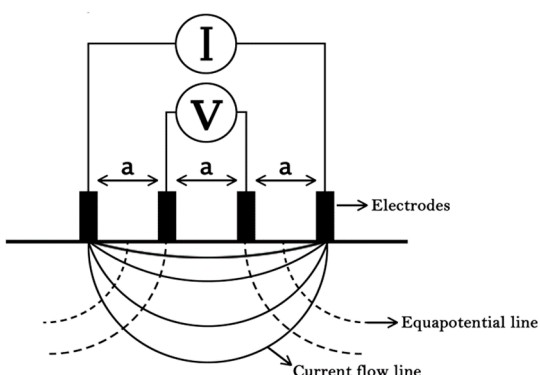

**Figure 3.** Surface electrical resistivity measurement by Wenner method.

## 3. Results and Discussion

### 3.1. Strength Characteristics

Table 4 shows the measurement results of the compressive and flexural strengths of the four concrete mixes used in this study at different days of age. Clearly, the OPC concrete mix exhibited a relatively higher compressive strength at an early age (7 days), but the compressive strength of concrete containing WS and AS became relatively higher as the age increased. In particular, after 91 days, concrete containing AS (WS35AS05 and WS30AS10) exhibited slightly higher compressive strength than OPC concrete and a similar strength as WS40 concrete. This appears to be because of the influence of large quantities of larnite ($\beta$-$Ca_2O_4Si$) present in AS [10].

**Table 4.** Compressive and flexural strengths of concrete.

| Mix. Code | Compressive Strength (MPa) | | | Flexural Strength (MPa) | | |
|---|---|---|---|---|---|---|
| | 7 d. | 28 d. | 91 d. | 7 d. | 28 d. | 91 d. |
| OPC | 25.0 | 33.1 | 36.1 | 4.7 | 6.1 | 7.2 |
| WS40 | 21.5 | 35.0 | 40.1 | 4.3 | 7.0 | 8.0 |
| WS35AS05 | 20.4 | 34.8 | 38.5 | 4.1 | 7.2 | 8.3 |
| WS30AS10 | 18.5 | 33.8 | 38.1 | 3.8 | 7.1 | 7.9 |

In addition, the concrete containing WS and AS exhibited higher flexural strengths than OPC concrete after 28 days. At 91 days, the flexural strength of OPC concrete was ~7.2 MPa, whereas that of WS40, WS35AS05, and WS30AS10 concrete was approximately 8.0, 8.3, and 7.9 MPa, respectively. This indicates that an excellent long-term flexural strength development was achieved owing to the latent hydraulic property of WS in blended cement concrete and the generation of $\beta$-$C_2S$ hydrates by AS [8,11].

Figures 4 and 5 show the ratio (%) of the compressive and flexural strengths of each concrete mix to those of OPC concrete, for examining the effect of substituting WS and AS on the strength characteristics of concrete. It is seen that the compressive and flexural strengths of concrete where OPC was substituted by WS and AS were higher than those of OPC concrete after 28 days.

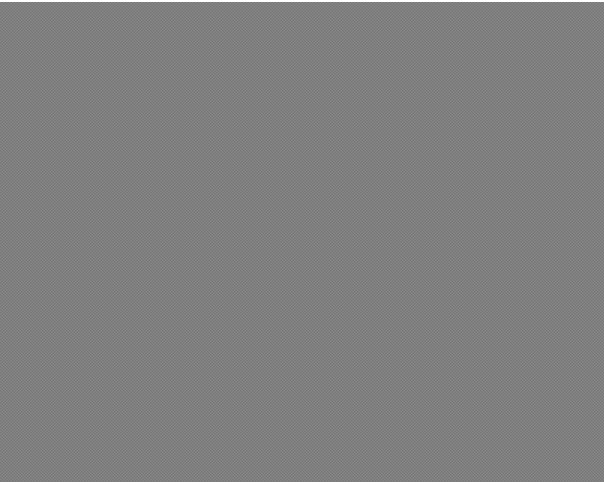

**Figure 4.** Compressive strength ratio of concrete according to the aging period.

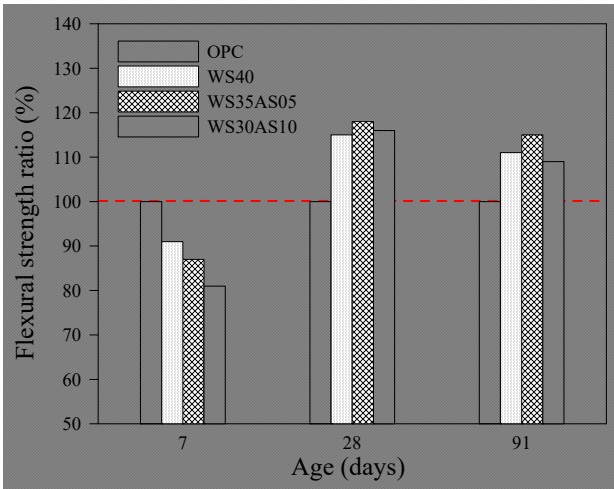

**Figure 5.** Flexural strength ratio of concrete according to the aging period.

### 3.2. Air-Void System of Concrete

The spacing factor ($\overline{L}$) in the air-void system is a function of the paste-to-air ratio (P/A) and specific surface area ($\alpha$), and it can be calculated as follows [18].

When P/A $\leq$ 4.33,

$$\overline{L} = \frac{P}{\alpha A} \tag{5}$$

When P/A > 4.33,

$$\overline{L} = \frac{3}{\alpha} \left[ 1.4 \left( \frac{P}{A} + 1 \right)^{1/3} - 1 \right] \tag{6}$$

In this study, the air content, specific surface area, and spacing factor of each concrete specimen after 28 days of age were measured using a QICAM digital camera (1.4 million, 1392 × 1040) and Image-Pro software, and the results are shown in Table 5. The experimental results showed that OPC concrete exhibited the largest spacing factor and the smallest specific surface area despite its highest air content. On the contrary, WS40 concrete had the smallest spacing factor and the largest specific surface area despite its relatively low air content. These results agree well with those of Sahin et al. [19].

**Table 5.** Air-void analyses results of hardened concrete.

| Mix. Code | Air Content (%) | Specific Surface Area (mm²/mm³) | Spacing Factor (mm) |
|---|---|---|---|
| OPC | 2.9 | 21.8 | 0.345 |
| WS40 | 2.3 | 39.1 | 0.264 |
| WS35AS05 | 2.6 | 34.8 | 0.284 |
| WS30AS10 | 2.1 | 33.5 | 0.288 |

### 3.3. Freeze–Thaw Resistance

It is known that in the concrete subjected to freeze–thaw cycles, microcracks are induced and concrete deterioration is accelerated, because the expansion of the water inside concrete due to freezing increases the tensile stress [20,21]. In this study, an exposure test was conducted according to ASTM C 666 to examine the freeze–thaw resistance of the four concrete mixes. The frost resistance of each concrete mix was evaluated by measuring the RDME, surface electrical resistivity, and compressive strength of concrete in predetermined freeze–thaw cycles.

After the dynamic modulus of elasticity was measured for each freeze–thaw cycle to evaluate the durability performance of the concrete mix subjected to freeze–thaw cycles, the RDME was obtained using Equation (1), which is plotted in Figure 6. The measurement results showed that the RDME of OPC concrete (reference mix) was significantly lower than those of the other three concrete mixes after 150 cycles, and reached ~55.4% after 240 cycles. Thereafter, it was not possible to measure the dynamic modulus of elasticity of OPC concrete because the specimen was disintegrated due to the excessive expansive pressure caused by freezing and thawing. The RDME of concrete with 30% WS and 10% AS became ~78.2% after 300 cycles. For WS40 and WS35AS05 concretes, the RDME values were larger than 88% after 300 freeze–thaw cycles, suggesting excellent frost resistances.

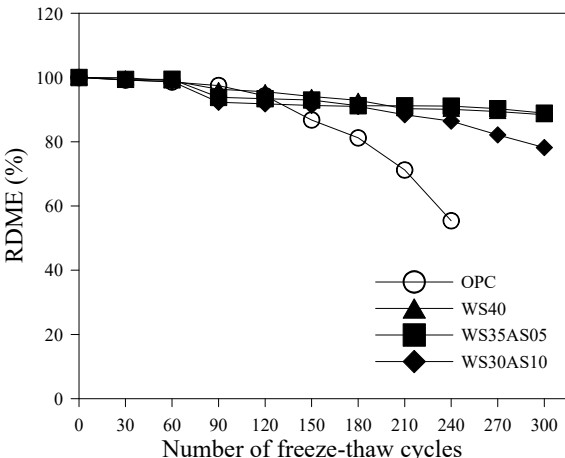

**Figure 6.** RDME values of concrete mixes exposed to freeze–thaw cycles.

In general, when concrete is exposed to a freeze–thaw environment, tensile stress is generated due to ice formation in the capillary pores. The entrained air in concrete relieves this stress caused by ice formation [22,23]. Therefore, reducing the capillary pores in concrete and introducing entrained air are very important for improving the freeze–thaw resistance of concrete. For slag-containing concrete that contains entrained air (approximately 6–7%), the freeze–thaw resistance increases because the capillary pores are reduced, and the expansion stress is relieved due to the latent hydraulic property [21,24,25]. Moreover, concrete containing AS exhibits the effect of relieving the tensile stress caused by ice formation, which is not observed in OPC concrete because the dense structure caused by the gradual generation of C-S-H reduces the capillary pores [26]. These experimental

results agree with those of Allahverdi et al. [21], who experimentally examined the frost resistance of concrete containing latent hydraulic binders (i.e., WS and AS).

Figure 7 shows the correlations between the DF of concrete and the results of air-void analyses (Table 5); the influence of the spacing factor and specific surface area on the DF can be clearly observed. In other words, the DF of the concrete subjected to freeze–thaw cycles increased as the spacing factor increased, but an opposite tendency was observed between DF and specific surface area [27]. For OPC concrete, which was significantly damaged by freeze–thaw cycles, in particular, DF was ~44.3% and the concrete exhibited a larger spacing factor and smaller specific surface area than those of the blended cement concrete.

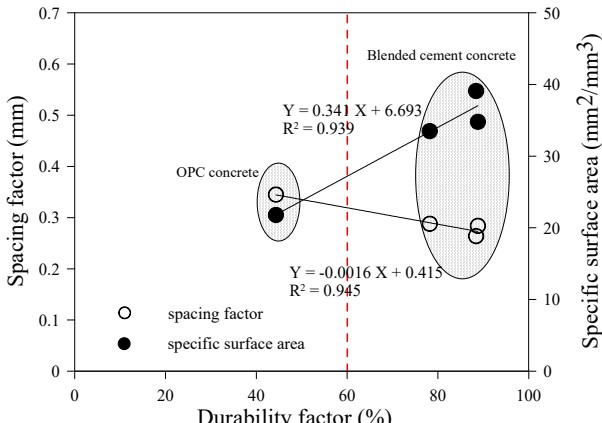

**Figure 7.** Relationship between the durability factor and air-void system of concrete.

Figure 8 shows the measured surface electrical resistivity of the concrete subjected to 0, 90, 210, and 300 freeze–thaw cycles. Regardless of the concrete mix, the surface electrical resistivity tended to decrease as the number of freeze–thaw cycles increased. Before exposure to the freeze–thaw environment, the surface electrical resistivity of OPC concrete was ~9.15 Kohm·cm, whereas that of the three blended cement concrete types was in the range 17.0–18.5 Kohm·cm, indicating that these structures were relatively denser. Figure 9 compares the surface electrical resistivity losses of all concrete mixes against the number of freeze–thaw cycles. Regardless of the number of freeze–thaw cycles, the resistivity of OPC concrete decreased most significantly. Although the surface electrical resistivity loss of OPC concrete was ~44.8% at 210 cycles, that of WS30AS10 concrete was ~24.5%, which was found to be the lowest.

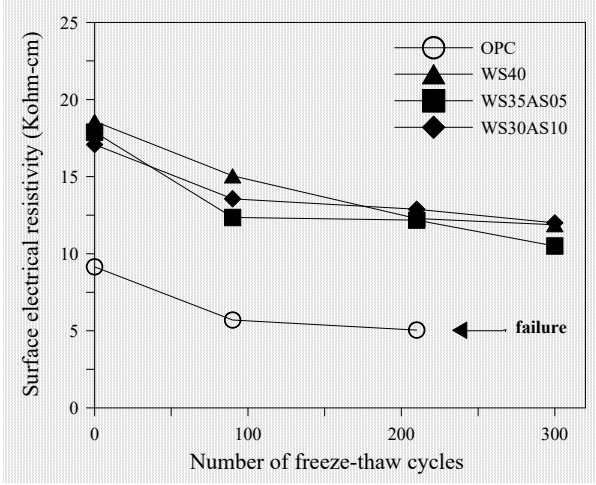

**Figure 8.** Surface electrical resistivity of concretes exposed to freeze–thaw cycles.

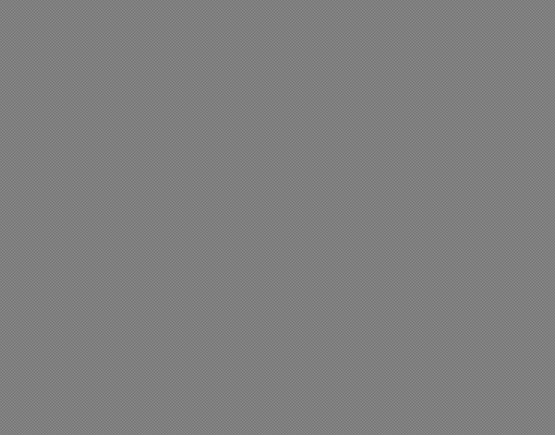

**Figure 9.** Comparison of surface electrical resistivity loss of concretes.

When concrete is exposed to the freeze–thaw environment, microcracks occur because of the increased porosity of the structure caused by the reduced density, resulting in reduction of the compressive strength [3]. Figure 10 shows the compressive strengths of the four concrete mixes measured before exposure to freeze–thaw cycles and at predetermined cycles. The compressive strength tendency was found to differ for each concrete mix. In particular, after 90 freeze–thaw cycles, the compressive strength of OPC concrete decreased compared with that before exposure to freeze–thaw cycles, but that of the remaining types containing blast furnace slag powder (WS and AS) increased slightly. This is because the increase in the strength of concrete caused by hydration was more significant than the decrease in strength caused by freeze–thaw cycles for the blended cement concrete—this tendency is similar to that observed by Allahverdi et al. [21]. Figure 11 shows the CLS of concrete at each cycle based on the compressive strength of concrete before exposure to freeze–thaw cycles. Regardless of the number of freeze–thaw cycles, the blended cement concrete exhibited negligible CLS (below 20%), indicating excellent frost resistance.

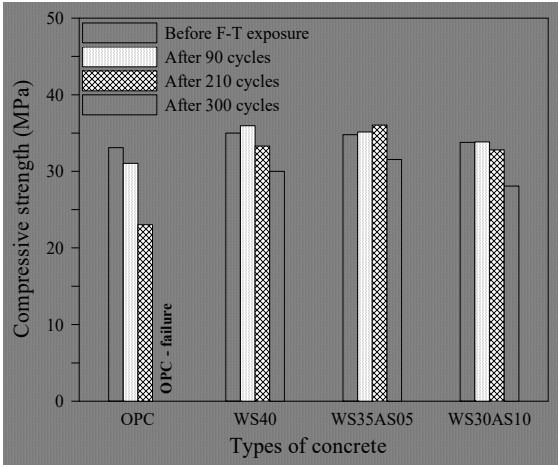

**Figure 10.** Compressive strength of concretes exposed to freeze–thaw cycles.

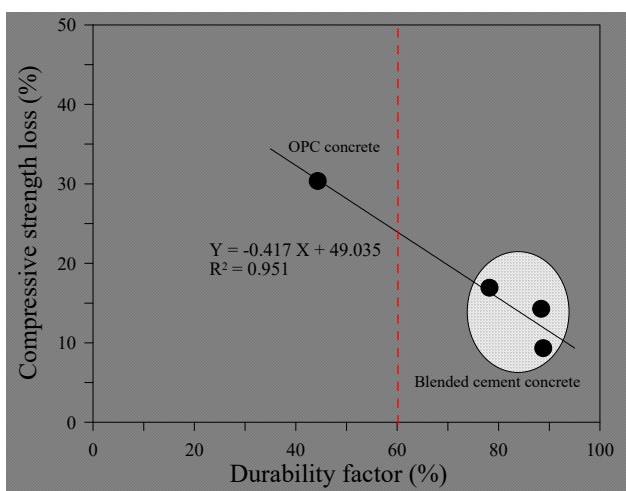

**Figure 11.** Comparison of compressive strength loss of concretes.

The relationship between durability factor (DF) and compressive strength loss (CSL) is presented in Figure 12. High correlation coefficient value (0.951) for two parameters were obtained, implying that the CSL of the concrete decreased as the DF increased. Overall, it can be said that the measurement of CLS may be a useful method to determine the durability of concrete under freeze–thaw cycles.

**Figure 12.** Relationship between DF and compressive strength loss of concretes.

In fact, the effect of WS on the durable performance of concrete under freeze–thaw cycles was reported by many researchers [28–30]. For example, Shen et al. studied that the permeability and water absorption of concrete with different replacement levels of WS exposed to a combined interaction of the fatigue load and freeze–thaw cycles, and established a model of the relationship between flexural strength and pore structure [29]. However, little studies carried out to verify influence of AS on the resistance of concrete against repeated freeze–thaw action. Results clearly demonstrated that if there is a proper mix proportion, the application of AS may improve the performance of concrete.

### 3.4. Microstructure Studies by SEM

The interfacial transition zone (ITZ), which is the boundary between the paste and aggregate, is the weakest part in the concrete structure, and the damages caused by freeze–thaw cycles, such as aggregate fallout, softening, and cracking, mainly occur in this region [31]. In this study, microstructural observations of the surface of the concrete specimens were performed after the 210 cycles of freeze–thaw test, which identified the ITZ, micropores, cracks due to freeze–thaw, and hydrates of concrete (Figure 13). The

OPC concrete subjected to freeze–thaw cycles had relatively less dense microstructures containing many micropores and microcracks, and very wide cracks (~1 μm) were observed near the ITZ, confirming that relatively high damages occurred due to freeze–thaw cycles.

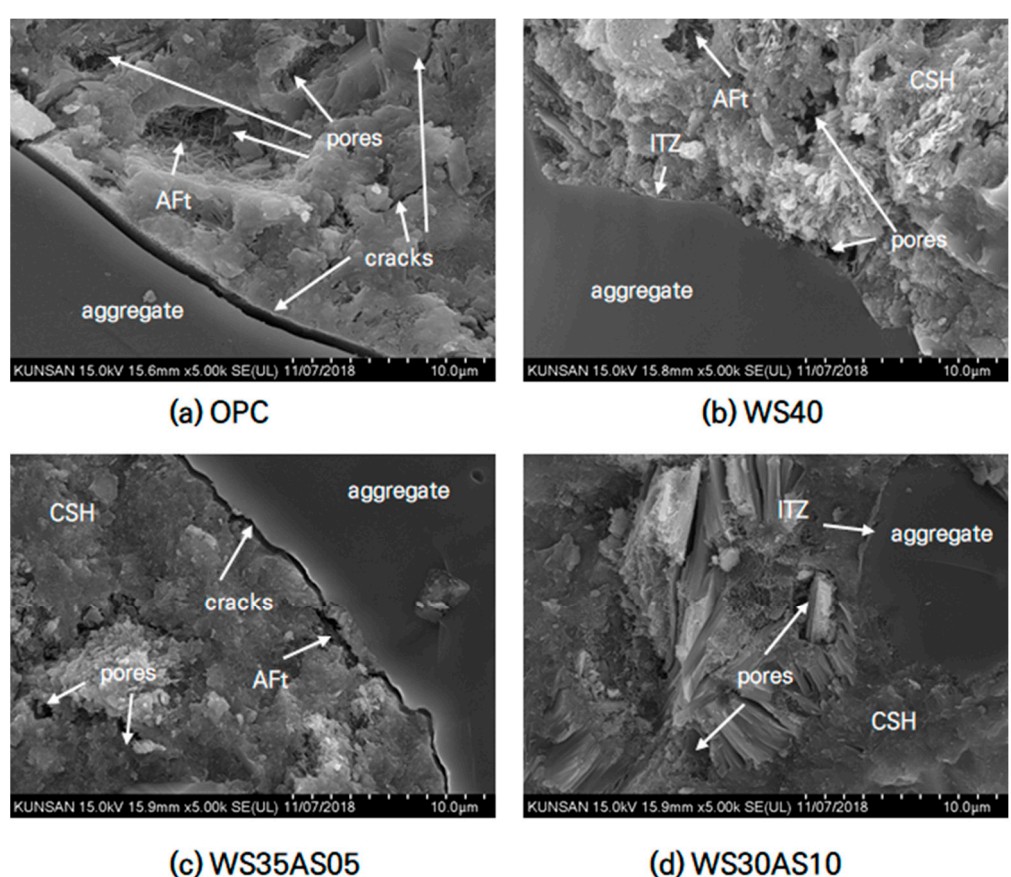

**Figure 13.** SEM images of concrete after the 210 cycles of freeze–thaw test.

On the contrary, a very dense microstructure and a few microcracks were observed near the ITZ for the WS40 concrete, confirming the generation of C-S-H in large quantities [8,32]. Fewer cracks and a denser ITZ were observed in WS30AS10 concrete. In WS35AS05 concrete, however, cracks (less than 0.5 μm) were observed near the ITZ, but distinct microcracks were rarely observed. Therefore, the blended cement concrete containing WS and AS based on the appropriate mix proportion is expected to exhibit excellent durability in regions experiencing very low temperatures.

## 4. Conclusions

In this study, the durability and microstructure of concrete containing water-cooled slag (WS) and air-cooled slag (AS) subjected to freeze–thaw cycles were experimentally examined. The following conclusions can be drawn.

(1) When the compressive and flexural strength characteristics of concrete were investigated, the strength of OPC concrete was found to be slightly higher at early ages, but the strength development of concrete containing AS (WS35AS05 and WS30AS10) was found to be superior to that of OPC concrete as the age increased, because of the influence of larnite, a β-$C_2S$ hydrate.

(2) When the air-void system of hardened concrete was measured, OPC concrete with high air content exhibited the largest spacing factor and the smallest specific surface area. On the contrary, WS40 concrete exhibited the smallest spacing factor despite its relatively low air content.

(3) The three blended cement concrete types subjected to 300 freeze–thaw cycles exhibited RDME values of ~88% or higher due to the influence of the latent hydraulic properties of AS and WS, suggesting relatively higher frost resistance than that of OPC concrete (~55.4% after 240 freeze–thaw cycles). The DF of the concrete subjected to freeze–thaw cycles exhibited an increasing tendency as the spacing factor increased, and the specific surface area decreased.

(4). The surface electrical resistivity of OPC concrete was found to be lower than that of blended cement concrete regardless of the number of freeze–thaw cycles. Although the surface electrical resistivity loss of OPC concrete was ~44.8% at 210 cycles, that of WS30AS10 concrete was ~24.5%. Furthermore, the compressive strength loss of concrete due to freeze–thaw cycles exhibited a tendency similar to that of the surface electrical resistivity, confirming a relatively superior frost resistance of blended cement concrete.

(5) When the microstructure of the concrete subjected to freeze–thaw cycles was analyzed, OPC concrete exhibited highly wide cracks near the ITZ, indicating relatively high damage caused by freeze–thaw cycles. On the contrary, WS40 concrete exhibited very dense microstructures near the ITZ, and concrete containing AS also exhibited excellent frost resistance with relatively small damages.

(6) Blended cement concrete containing WS and AS based on an appropriate mix proportion design will exhibit improved durability in regions experiencing harsh winters and very low temperatures. More studies are needed to determine the appropriate mix proportions and percentages of blended AS and WS.

**Author Contributions:** Conceptualization, S.-T.L.; methodology, S.-T.L. and S.-H.P.; software, S.-H.P.; validation, S.-T.L., D.-G.K. and J.-M.K.; formal analysis, D.-G.K. and J.-M.K.; investigation, S.-H.P. and D.-G.K.; resources, S.-T.L.; data curation, S.-T.L.; writing—original draft preparation, S.-T.L.; writing—review and editing, S.-T.L. and S.-H.P.; visualization, S.-T.L.; supervision, S.-T.L.; project administration, S.-T.L.; funding acquisition, S.-T.L. All authors have read and agreed to the published version of the manuscript.

**Funding:** This research was funded by the Ministry of Land, Infrastructure and Transport (MOLIT) and the Korea Agency for Infrastructure Technology Advancement (KAIA) for a project on "Development of anti-frost concrete pavement incorporating high volume air-cooled slag and its application (No. 21CTAP-C156959-02)".

**Institutional Review Board Statement:** Not applicable.

**Informed Consent Statement:** Not applicable.

**Data Availability Statement:** Not applicable.

**Conflicts of Interest:** The author declares no conflict of interest.

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
