# Peer review of "Effect of Freeze–Thaw Cycles on the Performance of Concrete Containing Water-Cooled and Air-Cooled Slag"

_applsci, doi:10.3390/app11167291_

Round 1
Reviewer 1 Report
Line 79. Please add more information about cement, 32.5 , 42.5, 52.5? N, S, R?
Line 82. Please add information about the SEM photo realization method
Table 1. By which method was the chemical composition tested? Please add the laboratory measurements of chemical compounds in table 1.
Table 2 Please add the full name F. M.
Preparation of two samples for testing the flexural strength does not allow for statistical processing of the results.
Line 148. Please apply the explanations directly next to the symbol.
Line 159. Please add information on what method the device is based on.
Line 197,208 Please add / correct reference numbers.
Line 202. How many samples were tested for air content?
Figures 6 and 7 are not included in the paper? The article is incomplete, which makes it impossible to verify.
The paper does not present the results of the flexural strength in the freeze-thaw cycles?
References to the results of other authors in the discussion are limited. The authors should separate Discussion part and to refer this in the Conclusions. Otherwise the paper looks like a technical report and not a scientific article.
Author Response
Response to Reviewer 1 Comments
Point 1: Line 79. Please add more information about cement, 32.5 , 42.5, 52.5? N, S, R?
Response 1: As the reviewer indicated, more detailed information about cement used in the study was added in the revised version.
Point 2: Line 82. Please add information about the SEM photo realization method.
Response 2: The SEM images of the binders were obtained using an XL-30 ESEM at the same magnification. Please refer to the revised version.
Point 3: Table 1. By which method was the chemical composition tested? Please add the laboratory measurements of chemical compounds in table 1.
Response 3: The binders had been produced by local plants, and data available on the binders were supplied by the manufacture of the binders, as shown in Table 1.
Point 4: Table 2 Please add the full name F. M.
Response 4: As the reviewer indicated, the term ‘F.M.’ was defined in Table 2.
Point 5: Preparation of two samples for testing the flexural strength does not allow for statistical processing of the results.
Response 5: Authors completely agree with the reviewer’s comment. According to ASTM C293, however, the flexural strength test with two samples produced from the same concrete batch can be conducted within a precision limit of 12%. Although it seems to be less statistical values, the results of two samples may stand for the flexural strength characteristics of the concretes. Authors would like to thank the reviewer for the valuable comment.
Point 6: Line 148. Please apply the explanations directly next to the symbol.
Response 6: As the reviewer indicated, the explanation was added next to the symbol. Please refer to the revised version.
Point 7: Line 159. Please add information on what method the device is based on.
Response 7: Some information of the device for surface electrical resistivity was added in the manuscript. Please see the revised version.
Point 8: Line 197,208 Please add / correct reference numbers.
Response 8: It is authors’ mistake. As the reviewer stated, the references 18 and 19 were corrected in the new version. Thanks for your kind indication.
Point 9: Line 202. How many samples were tested for air content?
Response 9: Actually, one sample per concrete mix was tested for air content. The centre part of the 28-day concrete samples for air content test was selected to avoid the effect of bleeding formed in the upper surface.
Point 10: Figures 6 and 7 are not included in the paper? The article is incomplete, which makes it impossible to verify.
Response 10: Please accept authors’ apology for the mistake. It was indeed an error on the manuscript not to have included Figure 6 and 7. Addition of the figures was done in the revised version.
Point 11: The paper does not present the results of the flexural strength in the freeze-thaw cycles?
Response 11: Unfortunately, within the scope of this study, flexural strength measurement of concretes under freeze-thaw cycles was not carried out. Surely, the test should be done to evaluate flexural performance of concrete against frost attack. In this work, the variations in RDME, surface electrical resistivity and compressive strength were examined to evaluate durability of concretes subjected to freeze-thaw environment. Instead, the correlation of compressive strength loss (CLS) and durability factor (DF) was added in the revised version, as shown in Figure 12. Thanks for the valuable comment.
Point 12: References to the results of other authors in the discussion are limited. The authors should separate Discussion part and to refer this in the Conclusions. Otherwise the paper looks like a technical report and not a scientific article.
Response 12: Authors deeply appreciate that the reviewer gave the valuable comments to enhance the quality of manuscript. The manuscript emphasizes that with an appropriate mix proportion, the utilization of WS and AS could be a promising concrete material for degraded concrete structures, especially in regions experiencing freezing temperatures. A lot of revisions were made to satisfy readers and researchers in revised version. Additionally, some references of other authors were added in the manuscript. Could you please review again to consider the possibility of publication ?. Author believes that the reviewer can find the revised manuscript with wide modifications.

Reviewer 2 Report
- Figures 6 and 7 are missing from the manuscripts.
- Do the authors have photos of the freeze thaw prisms after 210 cycles and 300 cycles of OPC concrete to see how they compare to those with blended WS and AS? It will good to have some good photos in the paper.
- Did the authors try to correlate reduction of dynamic modulus (DM) to loss of compressive strength for the number of cycles at which the compression tests were conducted? This information can be useful to researchers.
- Based on the author's research and the results, can the authors recommend the most effective percentage substitute AS and WS blends to replace OPC?
- Can the authors continue freeze and thaw cycels on those specimes with AS and WS that did not fail after 300 cycles and see how more cycles they can go. This information can be useful and improves the article.
- Paragraph after conclusion (5), I suggest to revise that paragraph as follows '..........blended cement concrete containing WS 338 and AS based on an appropriate mix proportion design will exhibit improved durability in regions experiencing harsh winters and very low temperatures. More studies are needed to determine the appropriate mix proportions and percentages of blended AS and WS.
